# Assessment of Blood Endothelial Cell Biomarkers in Women and Men with Abnormal Body Mass and Paroxysmal Atrial Fibrillation Based on CHA2DS2-VASC Score: A Retrospective Study

**DOI:** 10.3390/ijms26083627

**Published:** 2025-04-11

**Authors:** Wiesław Sikora, Dominika Kanikowska, Jan Budzianowski, Edyta Kawka, Rafał Rutkowski, Katarzyna Korybalska

**Affiliations:** 1Department of Pathophysiology, Poznan University of Medical Sciences, 60-806 Poznan, Poland; wsikora@ump.edu.pl (W.S.); ekawka@ump.edu.pl (E.K.); rrutkowski@ump.edu.pl (R.R.); koryb@ump.edu.pl (K.K.); 2Department of Cardiology with Internal Disease Subunit, Puszczykowo Hospital, 62-040 Puszczykowo, Poland; 3Department of Cardiology, Collegium Medicum, University of Zielona Góra, Nowa Sól Multidisciplinary Hospital, 67-100 Nowa Sol, Poland; jan.budzianowski@gmail.com

**Keywords:** paroxysmal atrial fibrillation, obesity, endothelial cell biomarkers, CHA2DS2-VASC score

## Abstract

Endothelial dysfunction (ED) promotes and maintains atrial fibrillation (AF). Using a CHA2DS2-VASc score in women and men with paroxysmal AF, we aimed to determine which patients’ ED would be more pronounced. We recruited 47 females and 48 males (mean BMI 31 kg/m^2^ and 30 kg/m^2^, respectively) with paroxysmal AF and abnormal body mass and divided them into those with low (F < 3; M < 2) and high (F ≥ 3; M ≥ 2) CHA2DS2-VASC score. The blood samples were taken before AF ablation. Using Elisa tests, we measured tissue plasminogen activator (t-PA), plasminogen activator inhibitor 1 (PAI-1), vascular cell adhesion molecule 1 (sVCAM-1), intercellular adhesion molecule (sICAM-1), von Willebrand factor (vWF), and thrombomodulin (sTM). ED was more pronounced in females, expressed by higher endothelial cell marker concentrations: sVCAM-1 and sTM in low scores and sICAM-1 in high scores, CHA2DS2-VASc. Females were characterized by postmenopausal status, higher risk of thrombosis, lower GFR, and more frequent treatment with antiarrhythmic drugs. In contrast, males have only higher suppression of tumorigenicity 2 (ST2). In conclusion, women with paroxysmal AF exhibited more pronounced ED compared to men, regardless of their CHA2DS2-VASc scores. The soluble pro-inflammatory adhesion molecules and thrombomodulin emerge as the most sensitive biomarkers of ED elevated in females.

## 1. Introduction

Atrial fibrillation (AF) is the most common, clinically significant disorder related to heart rhythm and has increased significantly worldwide [1]. It had a higher prevalence and incidence in older adults and more frequent occurrence in men than women [2]. A recent comprehensive review increasingly highlights sex differences in atrial fibrillation across various aspects of the disease, including its epidemiology, mechanisms, management, and outcomes [3]. The implications of these findings are significant, as previous studies have shown that women with AF generally experience worse outcomes, including higher rates of AF-related complications and treatment failures, compared to men with the condition [4,5]. Research by Lang et al. has shown that females with AF face a higher risk of stroke compared to males, and the strokes they experience tend to be more severe [6]. Studies indicate that the prevalence and outcomes of AF differ between the sexes [7]. These differences may be attributed to distinct pathomechanisms of endothelial dysfunction (ED) influenced by sex hormones [8,9]. Estrogens enhance endothelial function through various mechanisms: (1) reducing oxidative stress, (2) modulating the renin–angiotensin system, and (3) attenuating the cellular endothelin-1 system [10,11].

The pathophysiology of AF is complex and not wholly understood, involving multiple factors. It likely includes mechanisms such as endothelial dysfunction, inflammation, injury to vascular endothelial cells, vasoconstriction, and prothrombotic responses to vascular injury [12]. The endothelium is essential for maintaining vascular homeostasis, as it regulates vasodilation, thrombosis, inflammation, and permeability [13]. Endothelial cell dysfunction is linked to cardiovascular diseases and conditions like diabetes, hypertension, obesity, aging, atherosclerosis, and kidney function [14,15,16,17,18]. The role of ED in AF is not only limited to thromboembolism, but it is also involved in promoting and maintaining arrhythmias, predicting adverse outcomes, and the recurrence of AF following cardioversion and ablation therapy [19,20,21]. Corban et al. [20] found that patients with normal coronary endothelial function have a significantly lower incidence of AF during long-term follow-up than those with coronary endothelial dysfunction, despite having similar AF risk factors. Furthermore, Okawa et al. [21] recently analyzed endothelial cell function in patients with an incidence of cardiovascular events after AF ablation. Endothelial function was measured by assessing the RHI (reactive hyperemia index). They documented that almost 80% of enrolled patients had ED, and the RHI value was found to be associated with the CHA2DS2-VASc score. The 5-year incidence of cardiovascular (CV) events was higher among patients with ED than those without ED (12% vs. 6%), especially in those with higher CHA2DS2-VASc scores [21]. The relationship between vascular endothelial function, measured as flow-mediated dilatation (FMD), and CHA2DS2-VASc score displayed reduced FMD in proportion to the higher values of the CHA2DS2-VASc score in patients with paroxysmal AF [22].

Concerning these data, we assessed ED using six serum endothelial biomarkers (t-PA, PAI-1, sICAM-1, sVCAM-1, vWF, sTM) in women and men with abnormal body mass and paroxysmal AF scheduled for AF ablation.

We hypothesized that, in individuals with paroxysmal AF, endothelial dysfunction, as assessed by serum endothelial biomarkers, would vary based on sex and the CHA2DS2-VASc score. By categorizing patients according to their CHA2DS2-VASc scores, we aim to determine whether, even with appropriate treatment for paroxysmal AF, patients still exhibit endothelial dysfunction differently by sex.

## 2. Results

### 2.1. Females and Males with Paroxysmal AF (Table 1)

The recruited group of females (F) and males (M) with paroxysmal AF does not differ in age, BMI, EHRA classification, degree of inflammation, and platelet activation (sCD40L0) (Table 1). The only value distinguishing males from females is a higher concentration of ST2, a parameter associated with cardiac stress and fibrosis. ST2 is a strong predictor of heart failure and death in patients with atrial fibrillation [23]. Women are distinguished from men by a higher mean value of CHA2DS2-VASC, confirmed by a higher concentration of D-Dimers (within the normal range), which confirms a higher risk of thromboembolism in females compared to males (Table 1). Females have higher TSH (within the normal range) compared to males. However, no differences were shown in thyroid diseases, which are similar in both groups (Table 1). Women have a lower GFR value (below the range; the GFR value in men is also below the range). They are more often treated with antiarrhythmic drugs, which, together with an increased risk of thrombosis, their postmenopausal status, and obesity, might indicate their worse cardiovascular condition, reflected in increased parameters of endothelial function, sICAM-1, and soluble TM (Table 1). Soluble thrombomodulin is a biomarker of vascular endothelial cell damage. Although females expressed a higher level than men, its value is within the normal range observed in healthy people (3–15 ng/mL). Its higher level is elevated in patients with various diseases connected with ED (>20–50 ng/mL) [24,25].

**Table 1 ijms-26-03627-t001:** Baseline characteristics of females and males with paroxysmal AF. The unpaired data were analyzed using parametric (unpaired *t*-test) and nonparametric (Mann–Whitney) *t*-tests. Categorized data were analyzed with the Chi-square test. A *p*-value < 0.05 was considered significant.

Parameters	Females Paroxysmal AF *n* = 47	Males Paroxysmal AF *n* = 48	*p*
Age, years	64 (42–73)	62 (48–76)	0.0563
BMI, kg/m^2^	31.9 (23.3–42.5)	29.9 (22.2–37.4)	0.0757
BMI > 25 ≤ 45, *n* (%)	45 (96)	43 (95)	0.2505
Il-6, pg/mL	3.2 (2.0–11.4)	3.2 (1.9–11.5)	0.6044
EHRA 1	0	0	------
EHRA 2	24 (37)	25 (40)	0.9208
EHRA 3	20 (31)	20 (32)	0.9303
EHRA 4	3 (5)	4 (6)	0.9787
Mean CHA2DS2-VASC	2 (1–6)	1 (0–3)	<0.0001
Mean HAS-BLED	1 (0–3)	1 (0–3)	0.1042
% EF	59 (50–61)	60 (40–63)	0.8014
TnT, µg/L	0.007 (0.003–0.87)	0.008 (0.004–0.010)	0.1262
CK-MB, U/L	15 (10–37)	14 (7–46)	0.3511
SBP	130 (100–160)	125 (110–140)	0.0859
DBP	80 (52–95)	80 (55–110)	0.1848
D-Dimers mg/dL	0.18 (0.03–1.4)	0.14 (0.02–0.95)	0.0303
ST2	0.85 (0.50–5.0)	1.35 (0.46–4.0)	0.0118
TSH, µU/mL	1.60 (0.40–7.7)	1.25 (0.22–12.3)	0.0398
GFR, mL/min	66 (40–114)	79.5 (37–135)	<0.0001
sCD40L, pg/mL	558 (179–985)	498 (190–1020)	0.5895
Cholesterol, mg/dL	193 (132–290)	152 (116–269)	0.1184
Glucose, mg/dL	98 (87–127)	104 (80–138)	0.2254
Smoking, *n* (%)	3 (6)	2 (4)	0.6286
Endothelial cell markers			
t-PA, ng/mL	1.47 (0.67–3.1)	1.31 (0.71–2.1)	0.2713
PAI-1, ng/mL	18.0 (10.4–37.5)	17.4 (9.5–49.4)	0.8313
sVCAM-1, ng/mL	252.1 (70.1–601.3)	223.1 (74.3–650.9)	0.1155
sICAM-1, ng/mL	34.3 (16.7–295.9)	25.5 (9.3–320.2)	0.0076
vWF, ng/mL	1.9 (0.17–4.7)	2.1 (0.46–5.43)	0.2096
sTM, ng/mL	3.6 (2.3–5.7)	3.2 (0.94–7.9)	0.0263
Comorbidities and medications			
Dyslipidemia, *n* (%)	23 (49)	15(31)	0.0785
Hypertension, *n* (%)	33 (72)	31 (64)	0.5585
Heart Failure, *n* (%)	1 (2)	1 (2)	0.988
Thyroid Diseases, *n* (%)	17 (36)	10 (21)	0.0975
Statins, *n*(%)	22 (47)	24 (50)	0.7556
ACE inhibitor, *n* (%)	10 (21)	9 (19)	0.7582
VKA, *n* (%)	17 (36)	9(19)	0.0569
NOAC, *n* (%)	30 (64)	39 (81)	0.0569
Beta-blockers, *n* (%)	43 (91)	37 (77)	0.0542
ARB, *n* (%)	19 (40)	17 (35)	0.6149
CCB, *n* (%)	14 (30)	9 (19)	0.2093
Diuretics, *n* (%)	11 (23)	7(15)	0.2727
Antiarrhythmic, *n* (%)	26 (55)	12 (25)	0.0026

Abrreviation: BMI—body mass index; EHRA—European Heart Rhythm Association; CHA2DS2-VASc—Congestive heart failure, Hypertension, Age ≥75, Diabetes, Stroke, Vascular disease, Age 65–74, Sex (female); HAS-BLED—Hypertension, Abnormal renal/liver function, Stroke, Bleeding history or predisposition, Labile INR, Elderly (>65 years), Drugs/alcohol concomitantly; EF—ejection fraction; TnT—cardiac troponin T; CK-MB—creatine kinase MB, specific for the heart muscle; SBP—systolic blood pressure; DBP—diastolic blood pressure; ST-2—suppression of tumorigenicity 2 (potential biomarker associated with cardiac stress and fibrosis); TSH—thyroid stimulating hormone; GFR, glomerular filtration rate; sCD40L—soluble ligand CD40 (release from activated platelets); t-PA—tissue plasminogen activator; PAI-1—plasminogen activator inhibitor 1; sVCAM-1—soluble vascular cell adhesion molecule 1; sICAM-1—soluble intercellular adhesion molecule; vWF—von Willebrand factor; sTM—soluble thrombomodulin; ACE—angiotensin converting enzyme inhibitor; VKA, vitamin K anagonist; NOAC, non-vitamin K antagonist oral anticoagulants; ARB—angiotensin II receptor blocker.

### 2.2. Criteria for Division Based on the CHA2DS2-VASc Score for Females and Males

We divided our group of females and males according to the CHA2DS2-VASC score for those with a low CHA2DS2-VASC score (female < 3, male < 2) and those with a high CHA2DS2-VASC score (female ≥ 3, male ≥ 2) [21]. While the CHA2DS2-VASc score is widely used in both clinical practice and scientific research, recent recommendations suggest an alternative. The newer CHA2DS2-VA score, recommended from 2024, excludes gender as a factor. Champsi et al. [26] found that while clinical risk scores were only modest predictors of outcomes, the CHA2DS2-VA score (which omits gender) performed better than the CHA2DS2-VASc score for primary outcomes—such as all-cause mortality, ischemic stroke, and arterial thromboembolism—in a British cohort of 28,623 females and 50,228 males with atrial fibrillation (AF) recruited between 2005 and 2020. The authors concluded that removing gender from the risk calculation might simplify anticoagulant treatment decisions for AF patients.

A recent study by Konsta Teppo et al. compared the performance of the CHA2DS2-VA and CHA2DS2-VASc scores in stratifying stroke risk among Finnish AF patients (*n* = 144,879). Examining two time periods, 2007–2008 and 2017–2018, they noted that initially, when females had a higher stroke risk associated with AF than males, the CHA2DS2-VASc score showed greater predictive accuracy. However, this difference diminished over time. By the end of the study, when sex-related differences in stroke risk had largely disappeared, the CHA2DS2-VA score showed marginally better performance [27].

The two groups of females, divided into low and high CHA2DS2-VASC scores, do not differ in BMI, EHRA classification, degree of inflammation and platelet activation, endothelial cell markers, and many other parameters presented in Appendix A. Women with high CHA2DS2-VASC scores were older, had a higher risk of bleeding (higher mean HAS-BLED), and more of them suffered from hypertension (84% vs. 56%). The group with a lower CHA2DS2-VASC score had higher TSH, but it was still within the standard range (Appendix A).

The two groups of males, divided into low and high CHA2DS2-VASC scores, do not differ in BMI, EHRA classification, degree of inflammation, platelet activation (sCD40L), endothelial cell markers, and many others presented in Appendix A. Men with high CHA2DS2-VASC scores were older, had a higher risk of bleeding (higher mean HAS-BLED), and were more often treated with statins, VKA, and ARB than men with lower CHA2DS2-VASC scores (Appendix A). The group with a lower CHA2DS2-VASC score has higher TSH but is still in the standard range (Appendix A). The males with low CHA2DS2-VASC scores were more often treated with NOAK (92% vs. 69%) (Appendix A).

### 2.3. Comparison of Females and Males with Low CHA2DS2-VASC Score

Women and men with paroxysmal AF and lower CHA2DS2-VASC scores do not differ in age, BMI, EHRA classification, or degree of inflammation and platelet activation (Table 2). Females had higher TSH levels than males, but still in the standard range. A non-statistically confirmed trend (*p* = 0.0689) was shown that women are more often diagnosed with thyroid diseases (44%) compared to men (20%) (Table 2). Women had lower GFR values (below the standard range; the GFR value was also below the range in males), were frequently treated with beta-blockers (92% vs. 68%) and antiarrhythmic drugs (60% vs. 24%), which, together with abnormal body mass and their postmenopausal status, might predispose them to CVD, reflected in increased levels of endothelial cell parameters (sVCAM-1 and TM) compared with males with a low CHA2DS2-VASC score (Table 2).

### 2.4. Comparison of Females and Males with High CHA2DS2-VASC Score

Women and men with paroxysmal AF and higher CHA2DS2-VASC do not differ in BMI, EHRA classification, degree of inflammation, and platelet activation (Table 3). Women with higher CHA2DS2-VASC score were (i) older, (ii) had a higher risk of bleeding assessed by the HAS-BLED scale, (iii) higher SBP (above normal range), (iv) higher D-Dimer concentration, but still within normal range, (v) had higher cholesterol level (within normal range), and (vi) lower GFR (below normal range, similarly GFR below normal range was observed in men), than men with higher CHA2DS2-VASC score. The ST2 level is higher in men than women, suggesting slightly worse cardiac conditions in men. The higher value of sICAM-1, the only parameter of endothelial function that distinguishes these two groups, is observed in women. A partial explanation might be the fact that recognized factors of endothelial dysfunction, such as age, postmenopausal status, obesity, increased SBP, and higher risk of CV morbidity in patients with low GFR, were associated with women with higher CHA2DS2-VASC scores (Table 3).

### 2.5. Correlation Between the CHA2DS2-VASc Score and Endothelial Cell Biomarkers in Females and Males with Paroxysmal AF (Table 4)

No correlations were observed between the CHA2DS2-VASc score and the serum endothelial cell parameters in female or male patients with abnormal body mass and diagnosed paroxysmal AF (Table 4).
ijms-26-03627-t004_Table 4Table 4Correlation between CHA2DS2-VASc score and endothelial cell biomarkers in females and males with paroxysmal AF. The data were analyzed using a nonparametric Spearman correlation test.ParametersFemales with Paroxysmal AFMales with Paroxysmal AFr*p*r*p*t-PA−0.0020.98520.2730.0604PAI-1−0.0710.63120.0290.8410sVCAM-1−0.3460.05680.2470.0902sICAM-10.1500.3134−0.2020.1665vWF0.2080.16010.2650.0686TM0.1890.20190.2920.0440Abbreviation: CHA2DS2-VASc—Congestive heart failure, Hypertension, Age ≥75, Diabetes, Stroke, Vascular disease, Age 65–74, Sex (female); t-PA—tissue plasminogen activator; PAI-1—plasminogen activator inhibitor 1; sVCAM-1—soluble vascular cell adhesion molecule 1; sICAM-1—soluble intercellular adhesion molecule; vWF—von Willebrand factor; sTM—soluble thrombomodulin.


## 3. Discussion

We conducted a retrospective study to assess endothelial dysfunction by analyzing six serum biomarkers: t-PA, PAI-1, sICAM-1, sVCAM-1, vWF, and sTM. Our study focused on both women and men with paroxysmal AF and abnormal body mass, categorizing the participants based on their CHA2DS2-VASc scores. The objective was to investigate whether patients with paroxysmal AF and abnormal body mass, even when adequately treated, exhibited more significant ED based on their sex. The recruited group of female and male patients did not differ in age, BMI, EHRA classification, or levels of inflammation and platelet activation. However, when compared to males, females exhibited a higher risk of thromboembolism, as indicated by a higher average CHA2DS2-VASc score. Additionally, females were more frequently treated with antiarrhythmic drugs and had lower GFR values. These factors, combined with their postmenopausal status and obesity, suggest a poorer cardiovascular condition in females. This is reflected in elevated levels of ED markers, such as pro-inflammatory sICAM-1 and soluble thrombomodulin, a glycoprotein whose soluble form indicates damage to endothelial cells. However, our patients displayed sTM levels observed in healthy people; their elevated level is associated with ED [24,25].

Increased levels of circulating adhesion molecules such as sICAM-1, sVCAM-1, vWF, and sTM are considered markers of ED, but studies examining their relationship with AF are inconsistent [28,29,30]. Willeit et al. conducted a prospective cohort study, which included 49% female participants, over a 20-year follow-up period. They found a correlation with soluble VCAM-1 levels; however, none of the other 13 inflammatory markers measured were significantly associated with new-onset AF [30]. In contrast, Schnabel et al. analyzed 12 inflammatory markers, including IL-6, MCP-1, ICAM-1, CRP, sCD40L, MPO, and P-selectin, and found no relationship between these markers and the incidence of AF [29].

Longitudinal studies on endothelial dysfunction indicate that women experience more significant impairment than men [31,32]. Llauradó et al. studied the link between endothelial dysfunction and arterial stiffness in adults with type 1 diabetes. In this research, women exhibited higher serum concentrations of sICAM-1 and sVCAM-1 compared to male patients [31]. In another study, Remmelzwaal et al. investigated the sex-specific longitudinal and interpersonal associations between low-grade inflammation and endothelial dysfunction using echocardiographic measures in a general population. The findings showed that participating women were characterized by higher concentrations of sICAM-1 than male patients [32].

The differences in the levels of endothelial molecules ICAM-1 and VCAM-1 between women and men can be attributed to several factors. These include the effects of sex hormones, variations in the immune system, genetic and epigenetic differences, and lifestyle and environmental factors [11,33,34]. Estrogen, at higher concentrations in women, can influence the expression of molecules such as VCAM-1 and ICAM-1 [35]. Research indicates that estrogen and progesterone may modulate the levels of these molecules, which, in turn, affect inflammation and endothelial function [36,37]. While there is substantial literature highlighting the protective effects of sex hormones, particularly estrogen, against atherosclerosis, some ongoing controversies persist. Notably, postmenopausal hormone therapy has not consistently shown protective benefits in preventing cardiovascular disease [38]. Additionally, the regulation of ICAM-1 secretion may be influenced by distinct immunological responses in women compared to men. Inflammation may contribute to the sex differences observed in the expression of these endothelial molecules. A study by Medenwald et al. involving patients who underwent coronary angiography found that women had elevated IL-6 levels compared to men, suggesting differences in immune regulation depending on sex [39]. Shin et al. identified significant sex-specific differences in endothelial cells regarding pro-inflammatory, pro-oxidant, and angiogenic characteristics, suggesting a more vulnerable phenotype in females [40].

When analyzing females and males based on the CHA2DS2-VASC score, it is evident that individuals with higher scores tend to be older and at an increased risk of bleeding. Additionally, women are more frequently diagnosed with hypertension. At the same time, men are more likely to receive treatment with statins, vitamin K antagonists (VKAs), and angiotensin receptor blockers (ARBs) compared to those with lower CHA2DS2-VASC scores. No significant changes were observed in the measured serum endothelial cell markers. Among females and males with low CHA2DS2-VASC scores, it was noted that women had a GFR below the standard range. They also showed a higher prevalence of beta-blocker and antiarrhythmic drug usage. These factors, coupled with their postmenopausal status and abnormal weight, may indicate a poorer cardiovascular condition in women. This is further evidenced by elevated levels of endothelial cell markers, such as the pro-inflammatory inducible sVCAM-1 and the soluble form of thrombomodulin, compared to males with low CHA2DS2-VASC scores. In summary, women affected by ED are characterized by several factors, including postmenopausal status, obesity, lower GFR, older age, higher SBP, and more frequent use of beta-blockers and anti-arrhythmic medications. These characteristics suggest a worse cardiovascular status in women compared to men, which is reflected in their elevated serum levels of endothelial cell biomarkers. Moreover, markers indicating endothelial cell damage, such as sICAM and thrombomodulin, were consistently higher in women than in men, regardless of their CHA2DS2-VASc scores. Obesity [15], postmenopausal status [11], hypertension [14], low GFR [18], aging [16], and related inflammatory processes are significant factors contributing to ED. Cardiological treatments, which can enhance endothelial cell function, are essential for maintaining vascular health and preventing cardiovascular diseases. These treatments should be addressed to improve endothelial dysfunction, a critical factor in atherosclerosis and hypertension. Effective therapies include statins, ACE inhibitors, ARBs, diuretics, beta-blockers, and CCBs. These treatments improve nitric oxide bioavailability, reduce oxidative stress, promote endothelial repair, and decrease inflammation [41,42,43,44]. Antithrombotic treatment [45] in AF indirectly improves endothelial cell function by diminishing thrombin generation, which is a potent stimulus for platelets and the endothelium [45,46,47]. This treatment may also help explain the presence of soluble thrombomodulin (sTM) in our patients and the levels found in healthy individuals. Membrane thrombomodulin is a protein produced mainly by vascular endothelial cells that locally regulates blood coagulation by neutralizing thrombin and activated protein C (APC system). This mechanism prevents clot formation. Its sTM in the blood is recognized as a marker of endothelial damage, increasing during inflammatory processes and oxidative stress (e.g., sepsis, autoimmune disorders, DIC, atherosclerosis, hypertension, ACS, kidney failure, neoplasms, etc.) having limited ability to activate protein C [24,25]. Research suggests that women tend to have higher levels of thrombomodulin than men, which may be attributed to (i) hormone stimulation, (ii) a greater tendency to activate the fibrinolytic system and endothelial anticoagulant mechanisms, and (iii) generally better endothelial function [48,49]. After menopause, estrogen levels decline, which may reduce thrombomodulin levels in older women, eliminating differences between the sexes, but it seems that sTM levels rise during menopause due to the higher risk of cardiovascular diseases, causing ED [48,49,50]. The choice of treatment depends on the CHA2DS2-VASc score. The European guidelines recommend novel oral anticoagulants (NOACs) for patients with a CHA2DS2-VASc score of 1, with VKAs as an alternative. However, the assessment of ischemic stroke risk in AF patients should take individual characteristics into account [51].

In the present study, four of the parameters detected for ED are involved in various stages of hemostasis: t-PA, PAI-1, vWF, and TM. Notably, only TM was elevated in females (levels observed in healthy people), particularly those with low CHA2DS2-VASc scores. After four weeks of warfarin treatment, Frestone et al. reported no significant differences in vWF and sTM levels in the AF study population [28].

The limitation of our study was to detect only patients with paroxysmal AF, putting aside patients with persistent and long-standing AF. Furthermore, we only measured endothelial cell serum biomarkers commonly used in scientific research to assess ED. Although these biomarkers are easily detectable using commercially available ELISA tests, their effectiveness in evaluating ED is less robust compared to non-invasive methods used in clinical practice, pointing to FMD, pulse wave amplitude, RHI, or others [52].

In conclusion, females with abnormal body weight who suffer from paroxysmal AF exhibit worse endothelial cell conditions than males, regardless of their CHA2DS2-VASc score. The soluble pro-inflammatory adhesion molecules and thrombomodulin are the most sensitive biomarkers for identifying elevated endothelial dysfunction in women.

## 4. Materials and Methods

### 4.1. Ethics Statement

All patients who took part in the study signed a written consent. The study protocol was approved by the Medical Ethics Committee of the Poznań University of Medical Sciences (Decision number: 44/16, 81/17, 99/24).

### 4.2. Study Population

The study group comprised 171 patients (68 women: 43–73 years; 103 men: 38–76 years) with abnormal body weights (BMI > 25 ≤ 45 kg/m^2^) recruited between 2017 and 2020. Overweight and obesity were defined as a BMI of 25–30 kg/m^2^ and a BMI ≥ 30 kg/m^2^, respectively. The recruited patients were documented to have symptomatic paroxysmal, persistent, and long-standing atrial fibrillation (AF). Paroxysmal AF was defined as self-terminating AF episodes lasting <7 days, persistent AF that lasts longer than 7 days and could require cardioversion, while long-standing persistent AF occurred if continuous arrhythmia took at least ≥1 year.

Finally, in the retrospective study, we recruited 95 patients with abnormal body weight only with symptomatic paroxysmal AF (47 women: 43–73 years old with a mean BMI of 31 kg/m^2^ and 48 men: 48–76 years old with a mean BMI of 30 kg/m^2^) (Figure 1).

The exclusion criteria from this study were: no written consent to be included in the study, acute or chronic infection, diabetes, use of antibiotic therapy, malignancies, cardiac surgery in the previous 3 months, stroke, and acute coronary syndromes over the past 3 months, or exacerbation of heart failure.

All consecutive patients underwent a careful interview with an assessment of arrhythmia symptoms (EHRA scale), comorbidities, and drugs. A detailed physical examination (height, body mass, temperature, and blood pressure) was conducted. BMI index (was calculated by the researchers as weight in kilograms divided by the square of height in meters). CHA2DS2-VASC and HAS-BLED scores were also calculated [53]. The glomerular filtration ratio (GFR) was measured according to the CKD-EPI (Chronic Kidney Disease Epidemiology Collaboration) formula [54]. This formula is more accurate, especially for GFR > 60 mL/min/1.73 m^2^, and appears to be currently the best option for estimating GFR in the obese [55].

### 4.3. Biochemical Analyses

Venous blood was collected from all participants at hospital admission and taken on the day of AF ablation before the procedure. All routine biochemical analyses were performed immediately in a central hospital laboratory (hsTnT, CK-MB, cholesterol, glucose, TSH).

Additional serum samples were aliquoted and stored at −80 °C until assayed.

The concentration of the markers (vWF, sICAM-1, sVCAM-1, sTM, t-PA, PAI-1, sCD-40L, ST2) was measured by using DuoSet Immunoassay Development Kits (R&D Systems, Minneapolis, MN, USA) according to the manufacturer’s instructions. The hsIL-6 was measured using high-sensitivity R&D Immunoassay Development Kits (R&D Systems, Minneapolis, MN, USA). The sensitivity and the intra-assay average coefficient of variation between duplicates (% CV) were as follows: for vWF: 0.10 ng/mL, 5.7%; for sICAM-1: 0.02 ng/mL, 11.1%; for sVCAM-1: 0.01 ng/mL, 8.2%; for sTM: 0.03 ng/mL, 9.1%; for t-PA: 0.10 ng/mL, 8.8%; for: PAI-1: 0.14 ng/mL, 9.4%; for sCD-40L: 9.8 pg/mL, 7.7%; for ST2: 0.04 ng/mL, 8.4%; for hsIl-6: 0.04 pg/mL, 7.8%.

### 4.4. Statistical Analysis

Statistical analysis was performed using GraphPad Prism™ 6.00 (GraphPad Software Inc., La Jolla, CA, USA). The normal distribution of continuous variables was tested with the Shapiro–Wilk and D’Agostino-Pearson tests. The unpaired data were analyzed using parametric (unpaired *t*-test) and nonparametric (Mann–Whitney) *t*-tests. The correlation was tested using the Spearman test. Categorized data were analyzed with the Chi-square test. A *p*-value < 0.05 was considered significant. The data were expressed as median (min–max).

## Figures and Tables

**Figure 1 ijms-26-03627-f001:**
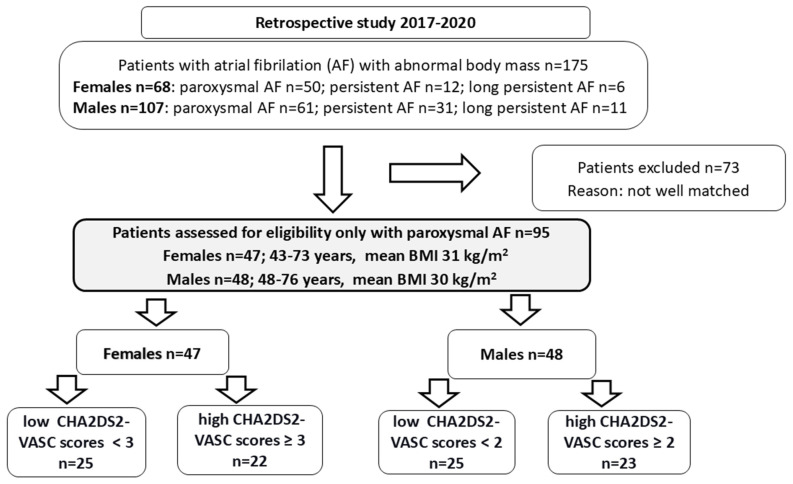
Patient eligibility for the study-flow diagram. Individuals with a history of AF.

**Table 2 ijms-26-03627-t002:** Characteristics of females and males with paroxysmal AF according to a low CHA2DS2-VASc score (female < 3; male < 2). The unpaired data were analyzed using parametric (unpaired *t*-test) and nonparametric (Mann–Whitney) *t*-tests. Categorized data were analyzed with the Chi-square test. A *p*-value < 0.05 was considered significant.

Parameters	Female CHA2DS2-VASC < 3 *n* = 25	Males CHA2DS2-VASC < 2 *n* = 25	*p*
Age, years	62 (43–73)	60 (48–69)	0.5242
BMI, kg/m^2^	30.9 (23.3–42.5)	30 (22–37)	0.2691
BMI > 25 ≤ 45, *n* (%)	23 (92)	22 (88)	0.6374
Il-6, pg/mL	3.2 (2.2–11.4)	3.2 (1.9–6.7)	0.3085
EHRA 1	0	0	------
EHRA 2	15 (60)	15 (60)	1.000
EHRA 3	10 (40)	9 (36)	0.7708
EHRA 4	0	1 (4)	0.3124
Mean HAS-BLED	1 (0–2) mean 0.76	1 (0–1) mean 0.68	0.7756
% EF	59 (50–60)	59 (50–63)	0.7806
TnT, µg/L	0.007 (0.003–0.084)	0.007 (0.004–0.024)	0.5071
CK-MB, U/L	15 (10–37)	14 (7–31)	0.2092
SBP	125 (100–160)	125 (110–140)	0.7147
DBP	75 (52–91)	80 (55–95)	0.2562
D-Dimers mg/dL	0.19 (0.03–1.36)	0.13 (0.03–0.66)	0.1235
ST2	1.2 (0.50–5.0)	1.4 (0.46–4.0)	0.1201
TSH, µU/mL	2.5 (0.66–7.7)	1.2 (0.22–12.3)	0.0063
GFR, mL/min	67 (44–114)	73 (50–135)	0.0320
sCD40L, pg/mL	559 (261–985)	512 (274–1020)	0.2293
Cholesterol, mg/dL	198 (160–267)	196 (116–269)	0.8972
Glucose, mg/dL	98 (88–127)	105 (80–138)	0.6754
Smoking, *n* (%)	2	3	0.6374
Endothelial cell markers			
t-PA, ng/mL	1.5 (0.67–3.1)	1.2 (0.71–2.1)	0.1453
PAI-1, ng/mL	18.0 (12.1–32.5)	17.5 (10.2–38.1)	0.8588
sVCAM-1, ng/mL	300.4 (111.4–601.3)	178.1 (74.3–510.4)	0.0055
sICAM-1, ng/mL	36.1 (16.7–261.9)	36.9 (9.3–108.1)	0.4212
vWF, ng/mL	1.8 (1.2–3.2)	1.8 (0.46–5.4)	0.9693
sTM, ng/mL	3.6 (2.4–5.1)	3.0 (1.1–4.9)	0.0150
Comorbidities and medications			
Dyslipidemia, *n* (%)	15 (60)	10 (40)	0.1573
Hypertension, *n* (%)	14 (56)	13 (52)	0.7766
Heart Failure, *n* (%)	1 (4)	0 (0)	0.3124
Thyroid Diseases, *n* (%)	11 (44)	5 (20)	0.0689
Statins, *n* (%)	10 (40)	7 (28)	0.3705
ACE inhibitor, *n* (%)	5 (20)	4 (16)	0.7128
VKA, *n* (%)	7 (28)	2 (9)	0.0657
NOAC, *n* (%)	18 (72)	23 (92)	0.0657
Beta-blockers, *n* (%)	23 (92)	17 (68)	0.0339
ARB, *n* (%)	9 (36)	7 (30)	0.5443
CCB, *n* (%)	6 (24)	3 (12)	0.2695
Diuretics, *n* (%)	5 (20)	4 (16)	0.7128
Antiarrhythmic, *n* (%)	15 (60)	6 (24)	0.0099

Abbreviation: BMI—body mass index; EHRA—European Heart Rhythm Association; CHA2DS2-VASc—Congestive heart failure, Hypertension, Age ≥75, Diabetes, Stroke, Vascular disease, Age 65–74, Sex (female); HAS-BLED—Hypertension, Abnormal renal/liver function, Stroke, Bleeding history or predisposition, Labile INR, Elderly (>65 years), Drugs/alcohol concomitantly; EF—ejection fraction; TnT—cardiac troponin T; CK-MB—creatine kinase MB, specific for the heart muscle; SBP—systolic blood pressure; DBP—diastolic blood pressure; ST-2—suppression of tumorigenicity 2 (potential biomarker associated with cardiac stress and fibrosis); TSH—thyroid stimulating hormone; GFR, glomerular filtration rate; sCD40L—soluble ligand CD40 (release from activated platelets); t-PA—tissue plasminogen activator; PAI-1—plasminogen activator inhibitor 1; sVCAM-1—soluble vascular cell adhesion molecule 1; sICAM-1—soluble intercellular adhesion molecule; vWF—von Willebrand. factor; sTM—soluble thrombomodulin; ACE—angiotensin-converting enzyme inhibitor; VKA, vitamin K antagonist; NOAC, non-vitamin K antagonist oral anticoagulants; ARB—angiotensin II receptor blocker.

**Table 3 ijms-26-03627-t003:** Characteristics of females and males with paroxysmal AF according to high CHA2DS2-VASc score (female ≥ 3; male ≥ 2). The unpaired data were analyzed using parametric (unpaired *t*-test) and nonparametric (Mann–Whitney) *t*-tests. Categorized data were analyzed with the Chi-square test. A *p*-value < 0.05 was considered significant.

Parameters	Females CHA2DS2-VASC ≥ 3 *n* = 22	Males CHA2DS2-VASC ≥ 2 *n* = 23	*p*
Age, years	68 (60–73)	66 (54–76)	0.0242
BMI, kg/m^2^	32.3 (25.3–36.7)	30 (22–36)	0.1022
BMI > 25 ≤ 45, *n* (%)	22 (100)	21(91)	0.1571
Il-6, pg/mL	3.4 (2.0–9.2)	3.2 (2.2–11.5)	0.8439
EHRA 1	0	0	------
EHRA 2	9 (41)	10 (43)	0.8615
EHRA 3	10 (45)	11 (48)	0.8734
EHRA 4	3 (14)	2 (9)	0.5981
Mean HAS-BLED	2 (1–3) mean 2.0	2 (0–3) mean 1.5	0.013
%EF	59 (50–61)	60 (40–60)	0.9982
TnT, µg/L	0.007 (0.003–0.87)	0.009 (0.005–0.022)	0.1286
CK-MB, U/L	15 (11–27)	17 (10–46)	0.6720
SBP	130 (110–160)	125 (110–140)	0.0210
DBP	80 (60–95)	80 (60–110)	0.5029
D-Dimers mg/dL	0.16 (0.07–0.54)	0.14 (0.02–0.95)	0.1839
ST2	0.78 (0.51–3.6)	1.3 (0.49–3.7)	0.0314
TSH, µU/mL	1.2 (0.40–6.3)	1.4 (0.58–3.2)	0.9684
GFR, mL/min	64 (40–91)	82 (37–113)	0.0006
sCD40L, pg/mL	512 (179–845)	494 (189–986)	0.6626
Cholesterol, mg/dL	177 (132–290)	138 (116–211)	0.0392
Glucose, mg/dL	100 (87–112)	98 (85–135)	0.8765
Smoking, *n* (%)	1	0	0.3011
Endothelial cell markers			
t-PA, ng/mL	1.5 (0.69–1.9)	1.4 (0.77–2.0)	0.9811
PAI-1, ng/mL	17.8 (10.4–37.5)	16.6 (9.5–49.4)	0.9958
VCAM, ng/mL	204.6 (70.1–590.9)	237.6 (102.1–690.9)	0.5471
ICAM, ng/mL	34.3 (19.3–295.9)	23.3 (9.6–320.2)	0.0016
vWF, ng/mL	1.9 (1.6–4.7)	2.7 (0.85–5.3)	0.0844
TM, ng/mL	3.6 (2.3–5.7)	3.3 (0.94–7.9)	0.7899
Comorbidities and medications			
Dyslipidemia, *n* (%)	8 (36)	5 (22)	0.2793
Hypertension, *n* (%)	19 (84)	18 (78)	0.4773
Heart Failure, *n* (%)	0 (0)	1 (4)	0.3226
Thyroid Diseases, *n* (%)	6 (27)	5 (22)	0.6659
Statins, *n* (%)	12 (55)	17 (74)	0.1749
ACE inhibitor, *n* (%)	5 (23)	7 (30)	0.5589
VKA, *n* (%)	10 (45)	7(30)	0.2989
NOAC, *n* (%)	12 (54)	16 (69)	0.2989
Beta-blockers, *n* (%)	19 (86)	20 (87)	0.9534
ARB, *n* (%)	10(45)	10(43)	0.8939
CCB, *n* (%)	3 (14)	6 (26)	0.2966
Diuretics, *n* (%)	5 (23)	3 (11)	0.3957
Antiarrhythmic, *n* (%)	11 (50)	7 (30)	0.1805

Abbreviation: BMI—body mass index; EHRA—European Heart Rhythm Association; CHA2DS2-VASc—Congestive heart failure, Hypertension, Age ≥75, Diabetes, Stroke, Vascular disease, Age 65–74, Sex (female); HAS-BLED—Hypertension, Abnormal renal/liver function, Stroke, Bleeding history or predisposition, Labile INR, Elderly (>65 years), Drugs/alcohol concomitantly; EF—ejection fraction; TnT—cardiac troponin T; CK-MB—creatine kinase MB, specific for the heart muscle; SBP—systolic blood pressure; DBP—diastolic blood pressure; ST-2—suppression of tumorigenicity 2 (potential biomarker associated with cardiac stress and fibrosis); TSH—thyroid stimulating hormone; GFR, glomerular filtration rate; sCD40L—soluble ligand CD40 (release from activated platelets); t-PA—tissue plasminogen activator; PAI-1—plasminogen activator inhibitor 1; sVCAM-1—soluble vascular cell adhesion molecule 1; sICAM-1—soluble intercellular adhesion molecule; vWF—von Willebrand. factor; TM—thrombomodulin; ACE—angiotensin-converting enzyme inhibitor; VKA, vitamin K antagonist; NOAC, non-vitamin K antagonist oral anticoagulants; ARB—angiotensin II receptor blocker.

## Data Availability

Data will be available on request.

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
