# Peer review of "Assessment of Blood Endothelial Cell Biomarkers in Women and Men with Abnormal Body Mass and Paroxysmal Atrial Fibrillation Based on CHA2DS2-VASC Score: A Retrospective Study"

_ijms, 2025, doi:10.3390/ijms26083627_

Round 1

Reviewer 1 Report

Comments and Suggestions for Authors

The topic is original to the field.
It addresses a specific gap in the field.
Compared with other published it adds new data. 
The conclusions are consistent with the evidence and arguments presented. They address the main question posed.

The references are appropriate.
The tables and figures are clear.

A retrospective study to assess endothelial dysfunction by analyzing six serum biomarkers: t-PA, PAI-1, sCAM-1, sVCAM-1, vWF and sTM focused on patients with paroxysmal AF and abnormal body mass. The results suggest a poorer cardiovascular condition in females with their postmenopausal status and obesity. Markers indicating endothelial cell damage were higher in women than men, regardless of their CHA2DS2-VASc scores.

Among four of the parameters detected for endothelial dysfunction involved in various stages of hemostasis, only soluble thrombomodulin was elevated in females, particularly those with low  CHA2DS2-VASc scores.

The soluble pro-inflammatory adhesion molecules and thrombomodulin are most sensitive biomarkers for identifying elevating endothelial dysfunction in women. 

The limitation of the study is effectiveness in evaluating endothelial dysfunction compared to non-invasive methods used in clinical practice.

------------------------------------

The topic is original to the field.
It addresses a specific gap in the field.
Compared with other published it adds new data. 
The conclusions are consistent with the evidence and arguments presented. They address the main question posed.
The references are appropriate.
The tables and figures are clear.

Author Response

Reviever 1- responses

The topic is original to the field.
It addresses a specific gap in the field.
Compared with other published it adds new data. 
The conclusions are consistent with the evidence and arguments presented. They address the main question posed.

The references are appropriate.
The tables and figures are clear.

 A retrospective study to assess endothelial dysfunction by analyzing six serum biomarkers: t-PA, PAI-1, sCAM-1, sVCAM-1, vWF and sTM focused on patients with paroxysmal AF and abnormal body mass. The results suggest a poorer cardiovascular condition in females with their postmenopausal status and obesity. Markers indicating endothelial cell damage were higher in women than men, regardless of their CHA2DS2-VASc scores.

Among four of the parameters detected for endothelial dysfunction involved in various stages of hemostasis, only soluble thrombomodulin was elevated in females, particularly those with low  CHA2DS2-VASc scores.

The soluble pro-inflammatory adhesion molecules and thrombomodulin are most sensitive biomarkers for identifying elevating endothelial dysfunction in women. 

The limitation of the study is effectiveness in evaluating endothelial dysfunction compared to non-invasive methods used in clinical practice.

Responses:

Dear Reviewer, thank you for your time and positive review of our work. Your opinion makes us happy and makes us feel that our research work is essential, for which we are very grateful.

Reviewer 2 Report

Comments and Suggestions for Authors

Unfortunately, since the new guideline for AF mention that there is a new score, respectively CHA2DS-VA score and not CHA2DS2-VASc (so the sex in not included anymore in this score), the entire manuscript needs to be rewritten accordingly.

Author Response

Reviewer 2 - responses

Unfortunately, since the new guideline for AF mention that there is a new score, respectively CHA2DS-VA score and not CHA2DS2-VASc (so the sex in not included anymore in this score), the entire manuscript needs to be rewritten accordingly.

Dear Reviewer, thank you very much for taking the time to review our manuscript.

All our improvements are in the manuscript in red.

  • Our study is retrospective, so we mentioned in the method section.

”Finally, in the retrospective study, we recruited 95 patients with abnormal body weight only with symptomatic paroxysmal AF (47 women: 43-73 years old with a mean BMI of 31 kg/m2  and 48 men:48-76 years old with a mean BMI of 30 kg/m2 ) (Fig.1).“

  • It is impossible to use a new score because, at the time we recruited patients (2017-2020), a CHA2DS2-VASc score was recommended to estimate the risk of stroke in patients with AF. According to this, patients were designated to the appropriate anticoagulant treatment, which implicates their clinical state, which we thoroughly analyze in our publication to assess the endothelial cell function. Nowadays, there are different recommendations (CHA2DS-VA score) that were not established between 2017-2020.

  • Now, we enriched our flow chart.
  • We add the information about data collection from 2017-2020 in the Method Section.

  • We also could point out that our study is retrospective in the title. It would now be entitled:

“Assessment of blood endothelial cell biomarkers in women and men with abnormal body mass and paroxysmal atrial fibrillation depending on CHA2DS2 – VASC score – A retrospective study.”

  • In our manuscript, in the Method Section, there was the following statement to be aware of a new recommendation published in 2024.

“While this scale is constantly used in clinical practice and scientific research, nowadays there are the new, recommendations [26].”

Reference 26:  Champsi, A.; Mobley, A.R.; Subramanian, A.; Nirantharakumar, K.; Wang, X.; Shukla, D.; Bunting, K.V.; Molgaard, I.; Dwight, J.; Arroyo, R.C.; Crijns, H.J.G.M.; Guasti, .;, Lettino, M.; Lumbers, R.T.; Maesen, B.; Rienstra, M.; Svennberg, E.; Tica, O.; Traykov, V.; Tzeis, S.; van, G.I.; Kotecha, D. Gender and contemporary risk of adverse events in atrial fibrillation. Eur Heart J 2024, 45, 3707-3717.

  • Now our statement is more precise:

“While this scale is widely used in clinical practice and scientific research, there are recent recommendations available.[26]. A recent study by Konsta Teppo et al. suggest that the CHA2DS2-VA score marginally outperforms the CHA2DS2-VASC in predicting ischemic stroke among contemporary patients with AF [27].

Reference 27.Teppo, K.; Lip, G.Y.H.; Airaksinen, K.E.J.; Halminen, O.; Haukka, J.; Putaala, J.; Mustonen, P.; Linna, M.; Hartikainen, J.; Lehto, M. Comparing CHA2DS2-VA and CHA2DS2-VASc scores for stroke risk stratification in patients with atrial fibrillation: a temporal trends analysis from the retrospective Finnish AntiCoagulation in Atrial Fibrillation (FinACAF) cohort. Lancet Reg Health Eur 2024, 10, 43:100967.

Round 2

Reviewer 2 Report

Comments and Suggestions for Authors

The authors responded to my question and elaborated their perspective. However, I recommend to add some more information about de difference between CHA2DS2-VA score and CHA2DS2-Vasc score, in order to consolidate the importance of this article.

Author Response

Dear reviewer,

Acording to your comment we improved imformation about the both scores

The authors responded to my question and elaborated their perspective. However, I recommend to add some more information about de difference between CHA2DS2-VA score and CHA2DS2-Vasc score, in order to consolidate the importance of this article.

Acording to your comment we have enhanced the information.

While the CHA2DS2-VASc score is widely used in both clinical practice and scientific research, recent recommendations suggest an alternative. The newer CHA2DS2-VA score, recommended from 2024, excludes gender as a factor. Champsi et al. [26] found that while clinical risk scores were only modest predictors of outcomes, the CHA2DS2-VA score (which omits gender) performed better than the CHA2DS2-VASc score for primary outcomes—such as all-cause mortality, ischemic stroke, and arterial thromboembolism—in a British cohort of 28,623 females and 50,228 males with atrial fibrillation (AF) recruited between 2005 and 2020. The authors concluded that removing gender from the risk calculation might simplify anticoagulant treatment decisions for AF patients.

A recent study by Konsta Teppo et al. compared the performance of the CHA2DS2-VA and CHA2DS2-VASc scores in stratifying stroke risk among Finnish AF patients (n=144,879). Examining two time periods, 2007–2008 and 2017–2018, they noted that initially, when females had a higher stroke risk associated with AF than males, the CHA2DS2-VASc score showed greater predictive accuracy. However, this difference diminished over time. By the end of the study, when sex-related differences in stroke risk had largely disappeared, the CHA2DS2-VA score showed marginally better performance [27].
